# Sexual Risk Behavior and Lifetime HIV Testing: The Role of Adverse Childhood Experiences

**DOI:** 10.3390/ijerph19074372

**Published:** 2022-04-05

**Authors:** Typhanye V. Dyer, Rodman E. Turpin, David J. Hawthorne, Vardhmaan Jain, Sonica Sayam, Mona Mittal

**Affiliations:** 1Epidemiology and Biostatistics, School of Public Health, University of Maryland, College Park, MD 20742, USA; rturpin@umd.edu (R.E.T.); ssayam@terpmail.umd.edu (S.S.); 2Behavioral and Community Health, School of Public Health, University of Maryland, College Park, MD 20742, USA; davidjh@umd.edu; 3Cleveland Clinic, Cleveland, OH 44195, USA; jainv2@ccf.org; 4Department of Family Science, School of Public Health, University of Maryland, College Park, MD 20742, USA; mmittal@umd.edu

**Keywords:** HIV, HIV prevention, HIV risk behaviors, HIV testing, adverse childhood experiences (ACEs)

## Abstract

Despite the success of HIV prevention drugs such as PrEP, HIV incident transmission rates remain a significant problem in the United States. A life-course perspective, including experiences of childhood adversity, may be useful in addressing the HIV epidemic. This paper used 2019 BRFSS data to elucidate the role that childhood adversity plays in the relationship between HIV risk and HIV testing. Participants (*n* = 58,258) completed self-report measures of HIV risk behaviors, HIV testing, and adverse childhood experiences (ACEs). The median number ACEs in the sample was 1, with verbal abuse (33.9%), and parental separation (31.3%) being the most common ACEs reported. Bivariate findings showed that all ACEs were associated with increased HIV risk and testing. However, increased risk was not correlated with increased HIV testing, with the highest incongruence related to mental health problems of household member (53.48%). While both self-reported HIV risk and ACEs were positively associated with HIV testing, their interaction had a negative association with testing (aPR = 0.51, 95%CI 0.42, 0.62). The results highlight the need for targeted HIV prevention strategies for at-risk individuals with a history of childhood adversity.

## 1. Introduction

It has been 40 years since the first cases of HIV were reported in the United States, and while efforts to reduce transmission have been successful with the advent of biomedical interventions, such as Truvada, Descovy, and Biktarvy for PrEP, incident transmission rates remain relatively stable [1]. In the United States, approximately 1.2 million people are living with HIV/AIDS, while over 36,000 individuals are acquiring HIV each year [2].

Risk factors for HIV transmission also remain stable, and include sharing of needles and syringes, having condomless intercourse, exchanging of sex for drugs, money, or other goods, and having other sexually transmitted infections [3,4,5]. These behavioral risk factors are additionally associated with a range of psychosocial sequelae, such as depression and suicidal ideation [6,7,8], psychological distress [9,10], alcohol and substance use and abuse disorders [11,12,13], disrupted social support [14,15,16], and experiences with violence [17]. Despite the extensive literature, which details HIV related risk behaviors and their synergistic impact on health outcomes, new perspectives that include a life-course perspective, are critical to ending the HIV epidemic. The application of a life-course lens can also potentially curb the onset of engaging in risk behaviors as early as adolescence, when exposure to adverse experiences that set individuals and communities on a trajectory for poorer health outcomes begin to take shape.

Adverse Childhood Experiences (ACEs) are potentially traumatic events that occur in childhood. ACEs can include experiencing abuse, witnessing violence, growing up in a family with mental health or substance use problems, and experiencing family instability due to divorce, parental separation, or incarceration. Ambient and often toxic stress from ACEs can alter brain development and affect how the body responds to stress, creating a bi-directional risk environment for poor health outcomes [18]. Large-scale studies in the United States, for example, demonstrate that the majority of the population has experienced at least one form of ACEs before they reached 18 years of age [19,20]. Additionally, ACEs have been linked to alteration of the neurobiological stress–response systems [21,22], resulting in negative physical health outcomes, such as cancer [23,24], respiratory problems, such as asthma [25,26] and chronic obstructive pulmonary disease [27,28], as well as premature mortality (Brown et al., 2009). ACEs are also linked with poor mental health and substance use outcomes in the extant literature [29,30], both of which are linked to engagement in HIV-related behaviors, both risk and preventative behaviors [31,32]. There is little literature on the relationship between ACEs, sexual risk behaviors, and engagement in HIV-related care outcomes, including testing.

### 1.1. ACES and HIV Risk

While ACEs and their impact on mental and behavioral health is well established [33,34], their association with HIV-related behaviors is less well understood [35,36,37]. A study by Feng et al. [36] found that ACEs were positively associated with HIV-related risk factors. Additionally, Hillis and colleagues [37] found a dose–response relationship such that those with one ACE experience had a slightly elevated likelihood of engaging in HIV related risk behaviors, while those with 4–5 or 6–7 such experiences had substantially elevated odds, including high numbers of sexual partners. Among individuals with a history of ACEs, risky sexual behavior may represent some attempt to achieve intimate interpersonal connections and intimacy. Extant studies have also explored the influence of ACEs on HIV-related behavioral risk in multiple populations, have examined gender and age differences, highlighting differential patterns of risk along gender and age characteristics [36]. The literature, while highlighting targets for interventions, has failed to address the synergistic effects of ACEs on HIV-related risk. A syndemic framework to guide our understanding of differential clustering of ACEs would allow for more refinement of targeting interventions to reduce risk behaviors. None have examined the impact of ACEs class membership on elucidating the association between HIV-related risk and HIV testing, which is the first and most critical step in engaging someone in HIV-care, that is known to reduce HIV transmission [38].

### 1.2. ACEs and HIV Testing

The first step in engagement in the HIV care continua is HIV testing. Early diagnosis allows for early access to treatment and access to treatment is associated with a reduced likelihood of onward transmission [1], better response to antiretroviral treatment (ART), and reduced mortality and morbidity [2] and ultimately better quality of life and life expectancy [39]. Despite significant investment of resources to increase the uptake of and improve testing and the CDC recommendation that individuals at high risk to HIV should be tested annually, HIV testing rates continue to be suboptimal. It is estimated that nearly 13% of people in the United States are unaware of their HIV status and that nearly 40% of the new HIV infections are transmitted by people who do not know that they are HIV positive [38]. Extant literature has underscored myriad psychosocial (e.g., depression and substance use, HIV stigma, and discrimination) [39,40,41,42] and structural level factors of unemployment, lower income, economic hardships, incarceration, and limited access to health care/antiretroviral treatment (e.g., being uninsured or underinsured) as contributors to decreased HIV testing access in BMSM [43,44,45] and that have been negatively associated with HIV testing. At an interpersonal level, social networks, including support provided by one’s network members has also been cited in the literature as being related to testing behaviors. Hermanstyne and colleagues [46] highlight that social support has positive benefits for health by affecting a person’s coping mechanisms or increasing their engagement in HIV health-promoting behaviors, such as testing [47]. Furthermore, the impact of the increased depth of one’s social support network as corresponding with better overall sexual health, which can contribute to the success of social network strategies that target HIV testing in BMSM [48,49]. While findings such as these are critical in increasing uptake of HIV testing [50], they fail to take into account a life course perspective wherein exposure to negative adverse events in childhood and adolescence, which may actually shape one’s propensity for depression, substance use, and developing unsupportive networks.

Few studies have examined the association between childhood adversity and HIV testing. Previous findings suggest that HIV testing outcomes may depend on the type of childhood adversity experienced. For example, in their study of adolescents in Malawi, Kidman and Kohler [51] found that parental substance use and parental incarceration were significantly associated with increased HIV testing, while physical abuse was significantly associated with decreased HIV testing in adjusted models. This study found a significant association between childhood adversity and HIV testing, which conflicts with a more recent study of MSM, which found no significant association between childhood adversity and testing [52].

The purpose of our study is twofold: First, to test for an association between self-reported HIV risk and ever having been HIV tested. Second, to determine if this association is modified by ACEs. We hypothesize that not only will there be a positive association between HIV risk and ever having been HIV tested, but that this association will be significantly different across levels of ACEs. Given the association highlighted in the previous literature between ACEs and both our exposure and outcome variables (i.e., HIV risk behaviors and HIV testing, respectively), and the underscoring put forth by Kidman and Kohler that HIV testing behaviors may differ across differing levels of ACE exposure, we hypothesize that not only will there be a positive association between HIV risk and ever having been HIV tested, but that this association will be significantly different across levels of ACEs.

## 2. Materials and Methods

Data from the 2019 Behavioral Risk Factor Surveillance System (BRFSS) were analyzed to determine the association between self-reported HIV risk and having been HIV tested in lifetime, modified by ACEs. BRFSS is an annual random-digit-dialed cellular and landline telephone survey of approximately 400,000 adult participants from the 50 states and District of Columbia (DC) in the United States [53]. A complex, stratified, and clustered design is used to collect the weighted data from both landline and telephone survey. Our analysis for this study is limited to 2019 data since this was the only year when information on both sexual/gender identity and ACEs were collected. Additionally, only the interviews where modules including the variables of interest were used for this study.

### 2.1. Exposure Variable

The main exposure variable for this study was self-reported HIV risk. As a measure for HIV risk, participants were asked to identify if any of the following statements apply to them (Yes/No): “You have injected any drug other than those prescribed for you in the past year; You have been treated for a sexually transmitted disease or STD in the past year; You have given or received money or drugs in exchange for sex in the past year; You had anal sex without a condom in the past year; You had four or more sex partners in the past year.” Participants did not have to specify which statement is true for them.

### 2.2. Outcome Variable

The outcome variable for our study was ever having been tested for HIV (Yes/No) outside of blood donations.

### 2.3. Moderator Variable

Our moderator was ACEs measured using 11 items, such as “Did you live with anyone who was a problem drinker or alcoholic?” or “How often did anyone at least 5 years older than you or an adult, force you to have sex?” Most items had binary outcomes (Did occur/Did not occur); for those that did not, these were dichotomized for consistency of coding. These items had good internal consistency (Cronbach’s alpha = 0.85) so we combined them into an index for analyses. This index was scaled in percentage (0% to 100%) for ease of interpretability.

### 2.4. Covariates

Covariates included age (18–24, 25–34, 35–44, 45–54, 55–64, 65, and older), race (Non-Hispanic White, Non-Hispanic Black, Non-Hispanic Multiracial, Non-Hispanic Other Race, Hispanic), sexual and gender identity (Cisgender bisexual women, Cisgender bisexual men, Cisgender gay women, Cisgender gay men, Cisgender heterosexual women, Cisgender heterosexual men, Cisgender other sexual identity women, Cisgender other sexual identity men, Cisgender questioning women, Cisgender questioning men, Non-Binary, Transgender women, Transgender men), income (Less than $15,000, $15,000 to less than $20,000, $20,000 to less than $25,000, $25,000 to less than $35,000, $35,000 to less than $50,000, $50,000 to less than $75,000, $75,000 or more), education (Did not graduate High School, Graduated High School, Attended College or Technical School, Graduated from College or Technical School), depression (Ever diagnosed, Never diagnosed), and binge drinking in the past 30 days (Yes, No).

### 2.5. Analyses

We tested bivariate associations between ACEs measures and HIV risk using a chi-square test for binary/multi-categorical measures and a Cochran–Armitage trend test for ordinal measures. We also reported proportions of each exposure, outcome, and covariate across HIV risk. We also tested associations between individual ACEs, as well as our ACEs index, and both HIV risk and HIV testing. We also examined congruence between the difference in HIV risk associated with ACEs and the difference in HIV testing associated with ACEs. This was to determine if the increased HIV risk associated with ACEs was completely accounted for with increased HIV testing.

For multivariate analyses, we constructed modified Poisson regression models with robust standard errors to generate prevalence ratios reflecting the differences in having been HIV tested between those with and without self-reported HIV risk. We also tested for interactions between HIV risk and our ACEs index using interaction terms. For models with and without interactions, we include unadjusted models and models adjusted for age, race, income, education, sexual identity depression, and binge drinking. We analyzed these cross-sectional data using survey weights for BRFSS provided by the CDC to account for the stratified and cluster design of the BRFSS to ensure representativeness to the population of the United States. We also conducted post hoc regression analyses examining the relationship between sexual and gender identity and HIV risk. We included unadjusted estimates and estimates adjusted for the ACEs index, income, age, highest education level, race/ethnicity, depression, and binge drinking.

### 2.6. Missing Data

Almost all variables had less than 1% missingness. We imputed missing measures using maximum likelihood imputation, using non-missing exposures and covariates. This allows for utilization of the most available information when estimating imputed values. After imputation, our final analytic sample consisted of 58,258 participants.

### 2.7. Quality Assurance

We assessed all regression model terms for intercollinearity by measuring their variance inflation factor. We also we did not find evidence of influential outliers using leverages and Cook’s distances. All tests used a two-sided test of significance at alpha = 0.05. All analyses were conducted using SAS 9.4.

## 3. Results

### 3.1. Sample Characteristics

Our sample (*n* = 58,258) was approximately 69% Non-Hispanic White, 14% Non-Hispanic Black, and 12% Hispanic/Latino (Table 1). The majority of our samples were aged 45 years and older. The majority (58%) reported having attended or completing either college or technical school. The majority of our sample was cisgender and identified as heterosexual men (42%) and heterosexual women (52%); 1% identified as transgender or non-binary. Nearly one fifth (18%) of the sample reported a history of binge drinking in the past 30 days, 20% had been diagnosed with depression, and 43% reported being ever tested for HIV. About 4% of our sample reported HIV risk behavior. Every other factor measured was strongly (*p* < 0.001) associated with HIV risk, including sexual minority identity, lower income, lower education, depression, binge drinking, and having been HIV tested. Participants reporting HIV risk were also more likely to be Hispanic or Black, and more likely to be younger.

### 3.2. ACEs Bivariate Analyses

The median number of ACEs in the sample was 1 out of 11, 9% when scaled in percentage (Table 2). The most common ACEs were having a parent or adult in your home ever swear at you, insult you, or put you down (33.9%), having parents separated or divorced (31.3%), and having a parent or adult in your home ever hit, beat, kick, or physically hurt you in any way other than spanking (25.0%). While all ACEs were associated with both greater HIV risk and greater testing, much of the increased HIV risk was not correlated with increased testing. The greatest incongruence was observed with ACEs related to mental health and substance use of household member, such as living with anyone who was depressed, mentally ill, or suicidal (53.8% incongruence). Overall, approximately half of the increased HIV risk was not associated with increased HIV testing for these measures.

### 3.3. Regression Modeling

Overall, both HIV risk and ACEs were positively associated with having been HIV tested (Table 3). When examining their interaction however, there was a strong negative interaction between ACEs and HIV Risk (Interaction aPR = 0.51, 95% CI 0.42, 0.62). At a higher number of ACEs, the positive association between HIV risk and HIV testing (aPR 1.51, 95% CI 1.38, 1.65), was greatly attenuated (Calculated aPR = 0.77, 95% CI 0.61, 0.98). These findings are also evident in adjusted proportions of HIV testing across ACEs and HIV risk (Figure 1). Results were slightly attenuated, but consistent after adjustment for covariates. Depression, binge drinking, higher education, younger age, Black and Hispanic race/ethnicities, and multiple sexual and gender identities were associated with greater HIV testing as well.

### 3.4. Post Hoc Analysis

Examining relationships between sexual and gender identity and HIV risk (Appendix A) only 2 of the 12 sexual and gender identity groups had significantly different self-reported HIV risk compared to cisgender heterosexual men after adjusting for covariates: Cisgender gay men (PR = 2.92, 95%CI 2.21, 3.85), and cisgender heterosexual women (PR = 0.62, 95%CI 0.52, 0.73). Due to variance inflation limitations, cisgender questioning men were not included in analyses, slightly reducing sample size for these analyses (from *n* = 58,258 to *n* = 58,077).

## 4. Discussion

Using a large, nationally representative sample of adults in the United States, this study explored the relationship between HIV risk behaviors, ACEs, and HIV testing. Sexual and gender identity, race, low SES, poor mental health, substance use, and previous HIV testing were found to be strongly associated with HIV-related risk behaviors. The most prevalent ACEs reported were verbal abuse, parental separation or divorce, parental substance abuse, and physical abuse. These findings are similar to previous studies estimating the prevalence of ACEs among persons with HIV [54] and in the general population [55]. Consistent with national estimates in the United States [56], approximately 43% percent of participants reported having ever been tested for HIV.

Our findings confirm the relationship between self-reported HIV risk behavior and HIV testing [57,58] and support the gradient-based relationship between ACEs and HIV testing highlighted in previous research [51]. As expected, both HIV-related risk and ACEs were positively associated with HIV testing. However, the interaction between HIV-related risk and ACEs had a strong negative association with HIV testing in adjusted regression models. A significant attenuation of the positive association between HIV-related risk and HIV testing was found with higher numbers of reported ACEs. There are several mechanisms through which ACEs could potentially impact HIV testing. First, considering ACEs’ association with lower socioeconomic status, individuals may have difficulty accessing care, lack consistent care, or lack a primary care provider [59]. Secondly, experiences of trauma, including ACES, are highly prevalent in the general population and particularly among people at high-risk for HIV [60]. Individuals exposed to ACES report higher levels of mental health issues such as depression, anxiety, PTSD, and substance use [61,62]. Poor mental health can interfere with HIV testing and learning one’s HIV status [32,63]. Moreover, people who are vulnerable to experiencing traumatic events and subsequent mental health issues often face structural and neighborhood factors (e.g., poverty, low education, unstable housing, unsafe neighborhoods) and discrimination and stigma that make it more challenging to engage in HIV preventative care [32]. While the recognition of the role of trauma in HIV risk has influenced the development of a small body of trauma-informed HIV prevention interventions, this knowledge has not been integrated in the area of HIV testing.

Additionally, we found that at higher levels of ACEs, HIV risk was less associated with increased testing in adjusted models. Among the ACES with the most pronounced risk-testing incongruence (i.e., poor mental health, substance abuse, or incarceration of someone in the household, and experience of verbal/emotional violence), about half of the increased risk related to ACEs was not accounted for. The level of incongruence between ACE-related HIV risk and HIV testing is concerning. The association between HIV risk perception and HIV testing outcomes has shown inconsistent results in the scientific literature, although the bulk of the evidence supports a positive association [64]. These findings suggest that perceived or actual HIV risk may be insufficient in terms of motivating individuals with a history of childhood adversity, particularly those that experience household dysfunction and emotional abuse during childhood, to get tested for HIV. Growing up in households with unpredictable, inconsistent, and chaotic caregiving as well as being emotionally put down disrupts attachment with caregivers in childhood. Attachment-related ACEs have been associated with significant adverse health and social outcomes [65,66,67]. Prior research has also found that household dysfunction is associated with poor health behaviors such as smoking, obesity, and alcohol dependency as well as poor socio-economic achievement [68]. Our study extends these findings further and highlights the impact of household-related dysfunction and emotional abuse on engagement in health behaviors, such as HIV testing.

Our findings highlight the urgent need for new programs and interventions that target high risk groups, promote testing benefits, and encourage increased testing among individuals with a history of ACEs. Although scientists have identified the negative impact of trauma on HIV and other health outcomes, there remains a dearth of trauma-informed HIV care models [69]. New, trauma-informed training programs for health care providers may be needed to provide effective care and improve healthcare outcomes. Given the importance of testing to HIV prevention outcomes, more research is needed to unpack the relationship between HIV risk and HIV testing for individuals with a history of childhood adversity.

Furthermore, these findings highlight the need for mobile testing units, home HIV testing kits, and testing through local community-based organizations are all effective strategies to improve access to HIV testing. Efforts to increase insurance coverage and eradicate barriers to health care broadly are also vital for HIV prevention efforts. Future research should consider qualitative methods to unpack potential barriers to HIV testing for individuals with a history of ACEs.

This study has several limitations. First, it should be noted that the ACE measure used for this study does not include all the potential experiences of childhood adversity that may be relevant in determining its impact on health behavior and health outcomes. For instance, the ACE measure omits relevant experiences of childhood adversity such as childhood neglect, community violence, familial death, and living with a significantly ill parent or guardian. Researchers have consistently called for a more exhaustive measure of household adversity to include such experiences [70]. Furthermore, no information about the severity, timing, or duration of ACE exposure was collected. Secondly, this study used cross-sectional data which limits our ability to make causal claims. Future longitudinal studies may be needed to understand temporal associations between ACEs, HIV risk, and HIV testing. Lastly, the HIV risk measure in the BRFSS is binary response and does not indicate which specific risk behaviors were endorsed. Future research may seek to study specific risk behaviors (i.e., condomless intercourse, transactional sex) and its effect on HIV testing outcomes. Moreover, studies could examine the effect of multiple risk behaviors on HIV testing. Despite these limitations, this study has several strengths including the use of a large, nationally representative BRFSS sample, standardized measures, and ACE specific consideration of HIV testing and HIV risk incongruence. Note that while most sexual and gender identity groups reported similar adjusted proportions of HIV risk, cisgender heterosexual women reported significantly less HIV risk, and cisgender gay men reported significantly more. This may reflect differences in the risk behaviors captured using the aggregate HIV risk measure. We recommend future sexuality and gender-specific research examining the relationship between ACEs, HIV risk, and HIV testing using individual HIV risk behaviors, to further elucidate this relationship.

## 5. Conclusions

In conclusion, our findings add to the body of literature on the relationship between childhood adversity, sexual risk behaviors, and HIV outcomes and underscore the need for increased prevention, assessment, and consideration of exposure to childhood adversity in the HIV care continuum. Findings have implications in both public health research and practice, including the need for further investigation on the role of childhood adversity on HIV-related behaviors and outcomes. Findings from this study reflect the critical need for more evidence-based, tailored interventions for individuals with a history of childhood adversity. This study establishes a critical link between HIV risk and testing behaviors and ACEs. It was one of the first studies to examine the link between HIV risk behaviors and outcomes related to ACEs. Highlighting the need for reducing HIV risk behaviors among young adult and adolescent people as key to interrupting HIV transmission. For that reason, the results from this study highlight the importance of preventing exposure to ACEs throughout the life course as an important HIV prevention strategy. Findings also underscore the urgent need for additional efforts to reduce and prevent childhood maltreatment and strengthen parents’ ability to protect against their children’s exposure to harmful adverse events. Study findings also expose the need to intervene with people who are affected and impacted by ACEs as early in the life course, as possible. This intervening can occur at the clinical level for general practitioners of medicine, as well as therapists and other clinical practitioners.

## Figures and Tables

**Figure 1 ijerph-19-04372-f001:**
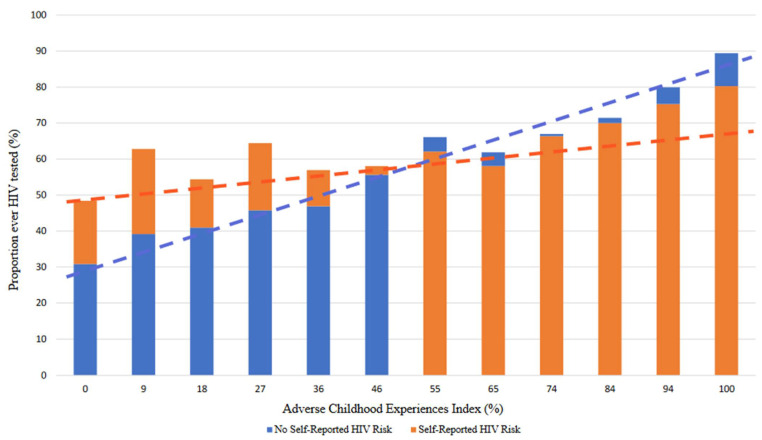
Adjusted Proportions of having been HIV tested stratified by self-reported HIV Risk and Adverse Childhood Experiences Index (*n* = 58,258). Larger bars are stacked behind smaller bars. Dotted lines represent trendlines. Significant (*p* < 0.001) interaction between the Adverse Childhood Experiences Index and self-reported HIV Risk based on regression modeling. Proportions adjusted for sexual and gender identity, age, race, income, education, depression, and binge drinking.

**Table 1 ijerph-19-04372-t001:** Proportions (%) of sample characteristics and associations between HIV Risk and demographics, sexual orientation/gender identity, depression, binge drinking, and HIV testing (*n* = 58,258).

	Total	No Self-Reported HIV Risk (*n* = 55,941, 96.0%)	Self-Reported HIV Risk (*n* = 2317, 4.0%)	*p* Value
**Sexual/Gender Identity** ^1^				<0.001
Cisgender Bisexual Women	1.40	1.81	7.58	
Cisgender Bisexual Men	0.69	0.76	3.48	
Cisgender Gay Women	0.57	0.67	0.61	
Cisgender Gay Men	0.77	0.73	7.14	
Cisgender Heterosexual Women	52.39	48.76	30.17	
Cisgender Heterosexual Men	41.56	43.92	47.59	
Cisgender Other Sexual Identity Women	0.59	0.56	0.87	
Cisgender Other Sexual Identity Men	0.37	0.48	0.82	
Cisgender Questioning Women	0.64	0.91	0.03	
Cisgender Questioning Men	0.31	0.41	0.33	
Non-Binary	0.41	0.64	0.47	
Transgender Women	0.14	0.19	0.59	
Transgender Men	0.14	0.16	0.32	
**Age** ^2^				<0.001
18–24	11.06	9.67	32.19	
25–34	14.44	13.44	29.46	
35–44	15.12	15.05	16.22	
45–54	15.93	16.24	11.3	
55–64	18.01	18.69	7.61	
65 and older	25.44	26.91	3.21	
**Race/Ethnicity** ^1^				<0.001
Non-Hispanic White	69.49	70.29	57.42	
Non-Hispanic Black	14.37	14.04	19.3	
Non-Hispanic Multiracial	0.73	0.72	0.89	
Non-Hispanic Other Race	2.21	2.28	1.18	
Hispanic	11.56	11.09	18.58	
**Income** ^2^				<0.001
Less than $15,000	4.97	4.93	5.57	
$15,000 to less than $20,000	8.14	8.01	10.16	
$20,000 to less than $25,000	10.58	10.61	10.22	
$25,000 to less than $35,000	12.01	11.96	12.74	
$35,000 to less than $50,000	14.15	13.94	17.34	
$50,000 to less than $75,000	15.8	15.78	16.08	
$75,000 or more	29.15	29.51	23.81	
**Education** ^2^				<0.001
Did not graduate High School	12.2	12.24	11.7	
Graduated High School	29.81	29.71	31.23	
Attended College or Technical School	32.01	31.55	39.13	
Graduated from College or Technical School	25.96	26.49	17.91	
**Ever Diagnosed with Depression** ^1^	20.3	19.33	34.98	<0.001
**Any Binge Drinking in the past 30 days** ^1^	17.62	16.16	39.72	<0.001
**Ever HIV Tested** ^1^	42.89	41.18	68.72	<0.001

^1^ Tested using chi-square test. ^2^ Tested using Cochran–Armitage trend test.

**Table 2 ijerph-19-04372-t002:** Proportions (%) of ACEs and median ACEs index stratified by self-reported HIV risk and ever having been HIV tested (*n* = 58,258).

	Total	Any Self-Reported HIV Risk		Any Self-Reported HIV Testing		Difference % in Risk Incongruent with Difference % in Testing ^1^
		No	Yes	No	Yes	
**Adverse Childhood Experiences** ^2^						
Did you live with anyone who was depressed, mentally ill, or suicidal?	18.16	16.96	36.24	14.31	23.28	53.48
Did you live with anyone who was a problem drinker or alcoholic?	24.11	23.27	36.82	20.77	28.55	42.58
Did you live with anyone who used illegal street drugs or who abused prescription medications?	11.33	10.22	28.15	7.65	16.24	52.09
Did you live with anyone who served time or was sentenced to serve time in a prison, jail, or other correctional facility?	9.15	8.19	23.62	5.71	13.73	48.02
Were your parents separated or divorced?	31.26	29.92	51.47	24.13	40.76	22.83
How often did your parents or adults in your home ever slap, hit, kick, punch or beat each other up?	17.56	16.91	27.44	14.08	22.19	22.98
Before age 18, how often did a parent or adult in your home ever hit, beat, kick, or physically hurt you in any way? Do not include spanking.	24.95	24.00	39.25	21.04	30.15	40.26
Did a parent or adult in your home ever swear at you, insult you, or put you down?	33.88	32.38	56.57	28.79	40.66	50.93
Did anyone at least 5 years older than you or an adult, ever touch you sexually?	11.74	11.04	22.4	8.28	16.35	28.96
Did anyone at least 5 years older than you or an adult, try to make you touch sexually?	8.86	8.25	18.1	5.95	12.74	31.07
Did anyone at least 5 years older than you or an adult, force you to have sex?	5.25	4.79	12.29	2.96	8.31	28.67
Median **Adverse Childhood Experiences** **Index** ^2^	9.01	9.01	27.27	9.01	18.18	33.33

^1^ This is calculated by dividing the difference in Self-Reported HIV testing by the difference in Self-Reported HIV Risk and taking the complement of this percentage (subtracting it from 100%). ^2^ All measures were statistically associated (*p* < 0.05) with both Self-Reported HIV Risk and Self-Reported HIV testing using a chi-square test (for individual items) and a Cochran–Armitage trend test (for the index).

**Table 3 ijerph-19-04372-t003:** Unadjusted and adjusted prevalence ratios (and 95% confidence intervals) for the association between HIV risk, ACEs, their interaction, covariates, and having ever been HIV tested (*n* = 58,258).

	No Interactions		Interactions	
	Unadjusted	Adjusted	Unadjusted	Adjusted
**Self-Reported HIV Risk**	**1.67 (1.58, 1.76)**	**1.24 (1.17, 1.31)**	**1.90 (1.74, 2.07)**	**1.51 (1.38, 1.65)**
ACEs Index			**2.77 (2.58, 2.97)**	**2.19 (2.03, 2.37)**
Self-Reported HIV Risk * ACEs Index ^1^			**0.44 (0.37, 0.54)**	**0.51 (0.42, 0.62)**
Sexual and Gender Identity				
Cisgender Bisexual Women		**0.74 (0.64, 0.85)**		**0.70 (0.61, 0.80)**
Cisgender Bisexual Men		**0.80 (0.68, 0.93)**		**0.79 (0.67 0.93)**
Cisgender Gay Women		1.00 (0.86, 1.17)		0.92 (0.78, 1.07)
Cisgender Heterosexual Women		**0.75 (0.67, 0.83)**		**0.77 (0.69, 0.85)**
Cisgender Heterosexual Men		**0.69 (0.62, 0.77)**		**0.72 (0.64, 0.80)**
Cisgender Other Sexual Identity Women		**0.63 (0.47, 0.83)**		**0.61 (0.47, 0.80)**
Cisgender Other Sexual Identity Men		**0.51 (0.32, 0.80)**		**0.53 (0.34 0.84)**
Cisgender Questioning Women		**0.63 (0.45, 0.88)**		**0.66 (0.47, 0.93)**
Cisgender Questioning Men		**0.37 (0.23, 0.61)**		**0.40 (0.24, 0.66)**
Non-Binary		**0.48 (0.33, 0.70)**		**0.52 (0.36, 0.75)**
Transgender Women		0.97 (0.73, 1.29)		0.95 (0.74, 1.22)
Transgender Men		0.77 (0.54, 1.08)		0.78 (0.54, 1.12)
Cisgender Gay Men		Reference		Reference
**Annual Household Income (Ordinal: Per $10,000 k)**		1.00 (0.99, 1.01)		0.99 (0.98, 1.00)
**Age (Ordinal: Per 10-years)**		**0.90 (0.89, 0.92)**		**0.91 (0.90, 0.92)**
**Highest Education Level (Ordinal: Per category ^2^)**		**1.06 (1.04, 1.08)**		**1.07 (1.05, 1.09)**
**Race/Ethnicity**				
Non-Hispanic Black		**1.75 (1.67, 1.82)**		**1.74 (1.67, 1.82)**
Non-Hispanic Multiracial		0.92 (0.76, 1.13)		0.98 (0.79, 1.20)
Non-Hispanic Other Race		**1.55 (1.45, 1.66)**		**1.56 (1.46, 1.67)**
Hispanic		**1.40 (1.28, 1.53)**		**1.33 (1.22, 1.46)**
Non-Hispanic White		Reference		Reference
**Ever Diagnosed with Depression**		**1.31 (1.26, 1.36)**		**1.18 (1.14, 1.23)**
**Binge Drinking past 30 days**		**1.09 (1.04, 1.14)**		**1.07 (1.02, 1.11)**

Significant (*p* < 0.05) results bolded. Adjusted for age, race, income, education, sexual identity depression, and binge drinking. ^1^ Interactions terms can be used to calculate the association between HIV risk and testing among those at the highest levels of the ACEs index (aPR = 0.77, 95%CI 0.61, 0.98). ^2^ Reflects a 1-category increase in the following: Did not graduate High School, Graduated High School, Attended College or Technical School, Graduated from College or Technical School. Ordinal analyses were used to maintain model integrity despite the large number of terms (>20 terms).*multiplication symbol.

## Data Availability

Data available in a publicly accessible repository that does not issue DOIs Publicly available datasets were analyzed in this study. This data can be found here: [https://www.cdc.gov/brfss/annual_data/annual_2019.html].

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
