# Peer review of "Sexual Risk Behavior and Lifetime HIV Testing: The Role of Adverse Childhood Experiences"

_ijerph, 2022, doi:10.3390/ijerph19074372_

Round 1

Reviewer 1 Report

Sexual risk behavior and Lifetime HIV Testing: The Role of Adverse Childhood Experiences

This is an interesting and important contribution to the field, but some changes would improve the overall quality of the article:

  1. Line 45: please include a definition of ACE.
  2. The study calls for a moderation analysis of ACE. I suggest authors to conduct this as it would deepen the complexity of results. Also, include a diagram explaining moderation results.
  3. Please include an implications’ section, discussing the impact of these results for public health policies and health education efforts.

Best wishes.

Author Response

Reviewer Comment 1: Line 45: please include a definition of ACE.

Author Response 1: Thank you for this comment, however unless there is a request for a more refined definition, in the original manuscript, Lines 45-49 read: “Adverse Childhood Experiences (ACEs) are potentially traumatic events that occur in childhood. ACEs can include experiencing abuse, witnessing violence, growing up in a family with mental health or substance use problems, and experiencing family instability due to divorce, parental separation, or incarceration. Ambient and often, toxic stress from ACEs can alter brain development and affect how the body responds to stress, creating a bi-directional risk environment for poor health outcomes.”  We feel this definition, which includes examples is sufficient for the reader to understand ACEs, as they have been described in prior literature.

Reviewer comment 2: The study calls for a moderation analysis of ACE. I suggest authors to conduct this as it would deepen the complexity of results. Also, include a diagram explaining moderation results.

Author Response 2: Thank you for this suggestion however, the effect modification analysis was one of the purposes of the manuscript and was already conducted. See lines 97-101, which read: “The purpose of our study is twofold: First, to test for an association between self-reported HIV risk and ever having been HIV tested. Second, to determine if this association is modified by ACEs. We hypothesize that not only will there be a positive association between HIV risk and ever having been HIV tested, but that this association will be significantly different across levels of ACEs.” Lines 124-130, which read: “Our moderator was ACEs measured using 11 items, such as “Did you live with anyone who was a problem drinker or alcoholic?” or “How often did anyone at least 5 years older than you or an adult, force you to have sex?” Most items had binary outcomes (Did occur / Did not occur); for those that did not, these were dichotomized for consistency of coding. These items had good internal consistency (Cronbach’s alpha = 0.85) so we combined them into an index for analyses. This index was scaled in percentage (0% to 100%) for ease of interpretability. Lines 155-158, which read: “We also tested for interactions between HIV risk and our ACEs index using interaction terms. For models with and without interactions, we include unadjusted models and models adjusted for age, race, income, education, sexual identity depression, and binge drinking.”  Finally, lines 200-208: “When examining their interaction however, there was a strong negative interaction between ACEs and HIV Risk (Interaction aPR=0.51, 95% CI 0.42, 0.62). At a higher number of ACEs, the positive association between HIV risk and HIV testing (aPR 1.51, 95% CI 1,38, 1.65), was greatly attenuated (Calculated aPR = 0.77, 95% 0.61, 0.98). These findings are also evident in adjusted proportions of HIV testing across ACEs and HIV risk (Figure 1). Results were slightly attenuated, but consistent after adjustment for covariates. Depression, binge drinking, higher education, younger age, Black and Hispanic race/ethnicities, and multiple sexual and gender identities were associated with greater HIV testing as well.

Our Table 3, also describes both our unmoderated and moderation analyses, showing the results of regression modeling using an interaction term for ACEs and HIV Risk, with HIV testing as the outcome. We describe these findings in the Results as follows:

“Overall, both HIV risk and ACEs were positively associated with having been HIV tested (Table 3). When examining their interaction however, there was a strong negative interaction between ACEs and HIV Risk (Interaction aPR=0.51, 95% CI 0.42, 0.62). At a higher number of ACEs, the positive association between HIV risk and HIV testing (aPR 1.51, 95% CI 1,38, 1.65), was greatly attenuated (Calculated aPR = 0.77, 95% 0.61, 0.98).”

Our Figure 1 shows the moderation results. These are described as well in the results:

“These findings are also evident in adjusted proportions of HIV testing across ACEs and HIV risk (Figure 1). Results were slightly attenuated, but consistent after adjustment for covariates.”

Reviewer Comment 3: Please include an implications’ section, discussing the impact of these results for public health policies and health education efforts.

Author Response 3: Thank you for this comment:  We have now included implications of our findings for the field of public health research and practice. Lines 292-306, now include implications for the study and reads: ”Findings have implications in both public health research and practice, including the need for further investigation on the role of childhood adversity on HIV-related behaviors and outcomes. Findings from this study reflect the critical need for more evidence-based, tailored interventions for individuals with a history of childhood adversity. This study establishes a critical link between HIV risk and testing behaviors and ACEs. It was one of the first studies to examine the link between HIV risk behaviors and outcomes related to ACEs. Highlighting the need for reducing HIV risk behaviors among young adult and adolescent people as key to interrupting HIV transmission. For that reason, the results from this study highlight the importance of preventing exposure to ACEs throughout the life course as an important HIV prevention strategy. Findings also underscore the urgent need for additional efforts to reduce and prevent childhood maltreatment and strengthen parents’ ability to protect against their children’s exposure to harmful adverse events. Study findings also expose the need to intervene with people who are affected and impacted by ACEs as early in the life course, as possible.

Reviewer 2 Report

Dear Authors 

Congratulations on your well written and presented  paper, and information warranted in the fight against HIV and AIDS. 

Author Response

Reviewer Comment 1: Congratulations on your well written and presented  paper, and information warranted in the fight against HIV and AIDS.

Author Response 1: Thank you so very much!

Reviewer 3 Report

The paper is logically argued, and the methodology is totally appropriate given the research questions asked.  The sample size is sound.  The analysis is well described.  Just a few minor issues.

  1.  You state: We hypothesize that not only will there be a positive association between HIV risk and ever having been HIV tested, but that this association will be significantly different across levels of ACEs.

can you introduction provide a stronger rationale and discussion to support putting forward this production?

2.  Why did you not include the variable substance use by the participants? 

3.  Can you suggest how your findings might influence GPs when discussing HIV testing with their patients?

Author Response

Reviewer Comment 1: You state: We hypothesize that not only will there be a positive association between HIV risk and ever having been HIV tested, but that this association will be significantly different across levels of ACEs.

can you introduction provide a stronger rationale and discussion to support putting forward this production?

Author Response 1: Absolutely.  We have now have amended the original manuscript to read: “Given the association highlighted in the previous literature between ACEs and both our exposure and outcome variables (i.e. HIV risk behaviors and HIV testing, respectively), and the underscoring put forth by Kidman and Kohler that HIV testing behaviors may differ across differing levels of ACE exposure, we hypothesize that not only will there be a positive association between HIV risk and ever having been HIV tested, but that this association will be significantly different across levels of ACEs.

Reviewer Comment 2: Why did you not include the variable substance use by the participants? 

Author Response 2: We had examined tobacco use, and marijuana use as potential covariates; this did not change our findings in any substantive way (less than a 5% change in estimates overall). Alcohol use was the only substance use measure in the BRFSS that substantially altered estimates, so it was the only one we included as a covariate. Note that alcohol use, tobacco use, and marijuana use are the only substance use measures in the BRFSS.

Reviewer Comment 3: Can you suggest how your findings might influence GPs when discussing HIV testing with their patients?

Author Response 3: While we appreciate this suggestion, we think it’s critical to make the distinction between implications for public health research and practice and clinical practice, given the scope of the findings, as well as the scope of the journal…being public health.  That being said, the implications now read: “Study findings also expose the need to intervene with people who are affected and impacted by ACEs as early in the life course, as possible. This intervening can occur at the clinical level for general practitioners of medicine, as well as therapists and other clinical practitioners.”

Reviewer 4 Report

Summary: This paper investigates the direct and interactive relationships among adverse child experiences (ACEs), HIV risk behaviors, and HIV testing. Performing bivariate and multivariable analyses, the authors showed that while both self-reported HIV risk  and ACEs were positively associated with HIV testing, their interaction had a negative association with testing, such that for people with more ACEs, the relationship between HIV risk and HIV testing was negative. Given the important role that childhood experiences play in creating conditions that make behavioral risk more likely as adults, this paper stands to make contribute greater nuance to our understanding of these early experiences with respect to HIV risk and prevention behavior. However, there are a few things that attenuate my enthusiasm for this paper that I would like to see addressed.

Comments:

 Lines 39-43: Long run on sentence - Break this up

Line 58: rephrase last part of sentence to: "to engagement in HIV related risk and prevention behaviors."

Section on “ACEs and HIV testing”:  I would like to see a summary of extant research about other known and, perhaps, more factors associated with HIV testing to establish what is currently known and the precise gap that ACEs can fill nature of the body of knowledge that this study aims to contribute to by shedding more light on the role of ACEs.

Regarding the measure of HIV Risk: Although I recognize that the agnostic measure of HIV risk may be a byproduct of the raw data (i.e., how the question was asked in the BRFSS), I have serious reservations about using it on such a heterogeneous population, as at least 2 of the 5 risk behaviors (i.e., survival/exchange sex and unprotected anal intercourse) are behaviors that are largely associated with certain subpopulations. If there is, indeed, no way to disaggregate the HIV risk measure (e.g., if binary indicators of each behavior are not available from the BRFSS), then I feel strongly that it is necessary to stratify the analysis (or do more posthoc analysis) to show: 1) where the significant differences actually lie between different gender & sexuality subgroups on HIV risk (more precisely than the authors already do), and 2) whether the relationships between HIV risk, HIV testing, and ACEs are any different by gender & sexuality subgroups.

Table 3: Please report additional variance explained with the addition of interaction terms.

Lines 268-276: The authors write that their findings highlight the need to address the social determinants of health. How do they do they support this? It’s not at all clear to me how they do that, unless you’re somehow situating ACEs in an SDOH framework. Not clear. Please connect your findings to an SDOH framework to support this claim or consider rethinking this claim altogether.

Author Response

Reviewer Comment 1: Lines 39-43: Long run on sentence - Break this up

Author Response 1: We have broken this sentence up.

Reviewer Comment 2: Line 58: rephrase last part of sentence to: "to engagement in HIV related risk and prevention behaviors."

Author Response 2:  We have edited the manuscript to reflect the suggested modification.

Reviewer Comment 3: Section on “ACEs and HIV testing”:  I would like to see a summary of extant research about other known and, perhaps, more factors associated with HIV testing to establish what is currently known and the precise gap that ACEs can fill nature of the body of knowledge that this study aims to contribute to by shedding more light on the role of ACEs.

Author Response 3: Thank you for pointing this out. This was an oversight on our part and the manuscript has been edited to reflect the gaps in the science around factors related to HIV testing.

Reviewer Comment 4: Regarding the measure of HIV Risk: Although I recognize that the agnostic measure of HIV risk may be a byproduct of the raw data (i.e., how the question was asked in the BRFSS), I have serious reservations about using it on such a heterogeneous population, as at least 2 of the 5 risk behaviors (i.e., survival/exchange sex and unprotected anal intercourse) are behaviors that are largely associated with certain subpopulations. If there is, indeed, no way to disaggregate the HIV risk measure (e.g., if binary indicators of each behavior are not available from the BRFSS), then I feel strongly that it is necessary to stratify the analysis (or do more posthoc analysis) to show: 1) where the significant differences actually lie between different gender & sexuality subgroups on HIV risk (more precisely than the authors already do), and 2) whether the relationships between HIV risk, HIV testing, and ACEs are any different by gender & sexuality subgroups.

Author Response 4: Yes, the aggregation of risk factors was how the question was asked in the BRFSS. We have added a supplementary table of prevalence ratios (and 95% confidence intervals) for the association between sexual and gender identity and self-reported HIV risk (Supplement 1). Notably, of the 12 sexual and gender identities examined, after adjusting for covariates, only 2 had significantly different self-reported HIV risk compared to cisgender heterosexual men: Cisgender gay men, and cisgender heterosexual women. We describe these findings in the following addition to the methods:

“We also conduct post-hoc regression analyses examining the relationship between sexual and gender identity and HIV risk. We include unadjusted estimates and estimates adjusted for the ACEs index, income, age, highest education level, race/ethnicity, depression, and binge drinking.”

We also describe findings in the results:

“Examining relationships between sexual and gender identity and HIV risk (Supplement 1), only 2 of the 12 sexual and gender identity groups had significantly different self-reported HIV risk compared to cisgender heterosexual men after adjusting for covariates: Cisgender gay men (PR=2.92, 95% CI 2.21, 3.85), and cisgender heterosexual women (PR=0.62, 95% CI 0.52, 0.73). Due to variance inflation limitations, cisgender questioning men were not included in analyses, slightly reducing sample size for these analyses (from n=58,258 to n=58,077).

We also discuss limitations and recommended future directions related to these findings in an addition to the discussion:

"Note that while most sexual and gender identity groups reported similar adjusted proportions of HIV risk, cisgender heterosexual women reported significantly less HIV risk, and cisgender gay men reported significantly more. This may reflect differences in the risk behaviors captured using the aggregate HIV risk measure. We recommend future sexuality and gender-specific research examining the relationship between ACEs, HIV risk, and HIV testing using individual HIV risk behaviors, to further elucidate this relationship.”

Regarding further substratification across HIV risk, testing, and ACES: Because we are already examining an interaction (the interaction between HIV Risk and ACEs, related to their associations with HIV testing) to further examine moderation of this interaction by sexual identity subgroup would require a three-way interaction term, which even for very large sample sizes is not feasible with reasonable power. Substratifying while maintaining interaction terms is not feasible for the same reasons.

Reviewer Comment 5: Table 3: Please report additional variance explained with the addition of interaction terms.

Author Response 5: We have added this to Table 3 underneath the models with interaction terms. We have also added a footnote describing that “the additional variance explained by adding interaction terms was calculated using the difference in an adjusted R squared, comparing each model to a model with the interaction term removed.”

Reviewer Comment 6: Lines 268-276: The authors write that their findings highlight the need to address the social determinants of health. How do they do they support this? It’s not at all clear to me how they do that, unless you’re somehow situating ACEs in an SDOH framework. Not clear. Please connect your findings to an SDOH framework to support this claim or consider rethinking this claim altogether.

Author Response 6: Thank you for catching this.  We have removed SDOH from the manuscript, as the results do not support this.

Round 2

Reviewer 4 Report

The authors have adequately addressed my comments. 

Minor edit required: lines 111-117 should be deleted; duplicative content

Author Response

THANK YOU!  Lines 111-117 have been deleted!

Typhanye